# Design of Cyclic Peptide-Based Nanospheres and the Delivery of siRNA

**DOI:** 10.3390/ijms232012071

**Published:** 2022-10-11

**Authors:** Junfeng Ke, Jingli Zhang, Junyang Li, Junqiu Liu, Shuwen Guan

**Affiliations:** 1School of Life Sciences, Jilin University, Changchun 130012, China; 2Engineering Laboratory for AIDS Vaccine, Jilin University, Changchun 130012, China; 3State Key laboratory of Supramolecular Structure and Materials, College of Chemistry, Jilin University, Changchun 130012, China

**Keywords:** cyclopeptides, covalent assembly, nanospheres, RNAi

## Abstract

In recent years, cyclic peptides have attracted much attention due to their chemical and enzymatic stability, low toxicity, and easy modification. In general, the self-assembled nanostructures of cyclic peptides tend to form nanotubes in a cyclic stacking manner through hydrogen bonding. However, studies exploring other assembly strategies are scarce. In this context, we proposed a new assembly strategy based on cyclic peptides with covalent self-assembly. Here, cyclic peptide-(DPDPDP) was rationally designed and used as a building block to construct new assemblies. With cyclo-(DP)_3_ as the structural unit and 2,2′-diamino-N-methyldiethylamine as the linker, positively charged nanospheres ((CP)_6_NS) based on cyclo-(DP)_3_ were successfully constructed by covalent self-assembly. We assessed their size and morphology by scanning electron microscopy (SEM), TEM, and DLS. (CP)_6_NS were found to have a strong positive charge, so they could bind to siRNA through electrostatic interactions. Confocal microscopy analysis and cell viability assays showed that (CP)_6_NS had high cellular internalization efficiency and low cytotoxicity. More importantly, real-time polymerase chain reaction (PCR) and flow cytometry analyses indicated that (CP)_6_NS-siRNA complexes potently inhibited gene expression and promoted tumor cell apoptosis. These results suggest that (CP)_6_NS may be a potential siRNA carrier for gene therapy.

## 1. Introduction

The occurrence and development of tumors are closely related to the abnormal expression of genes, and the down-regulation of abnormal genes is believed to inhibit the development of cancer [1,2,3]. siRNA is double-stranded and about 21 nucleotides long. It can accurately bind and act on specific mRNA through base pairing and induces degradation of mRNA, eventually leading to the failure of protein expression and the induction of apoptosis [4,5]. Therefore, therapy based on this strategy has great application potential for the treatment of some serious diseases, such as human severe combined immunodeficiency disease (SCID), genetic diseases, and cancer [6,7,8,9]. However, due to its large size and high negative charges, naked siRNA cannot pass through the cellular lipid bilayer by passive diffusion [10,11]. Furthermore, after endocytosis, it is easily degraded and cleared by enzymes [12,13]. Therefore, there is an urgent need to develop a carrier for the safe and effective delivery of siRNA to the target site.

With the development of nanotechnology, researchers have successfully designed and constructed a series of nanocarriers, including self-assembled nanoparticles, polymers, nanocapsules, and two-dimensional (2D) nanomaterials, which have been demonstrated to significantly improve siRNA transfection into cells [14,15,16,17,18,19,20,21]. However, some of them exhibit inherent biotoxicity and require multi-step preparations. Therefore, nanocarriers with good biocompatibility, high stability, and transfection efficiency need to be developed.

Biomolecule-based nanostructures have received extensive attention from researchers due to their good biocompatibility and low toxicity [22]. Proteins, nucleic acids, and peptides have been selected as the most important building blocks for assembly with other small molecules, generating a wide variety of complex nanostructures, which are chemically and physically stable and easy to modify so that they perform certain functions [23,24,25,26,27,28,29,30]. They are also diverse in structure and function, making them one of the most promising gene delivery vehicles.

Cyclic peptides have good biocompatibility and are easy to modify. They are promising gene delivery vehicles due to their high enzymatic stability, low toxicity, and ability to bypass endosomal uptake [22]. Compared to linear peptides, cyclic peptides have greater potential as stabilizers, a property that stems from their rigid structure and excellent hydrogen-bonding ability [31]. L-tryptophan and L-arginine-based cyclic peptides were reported to form nanostructures for siRNA delivery, and significant gene silencing was observed [31,32]. However, since the formed assemblies rely on non-covalent forces, they are not stable over long periods. Therefore, there is also a need to develop an efficient method to obtain stable nanocomponents with excellent siRNA delivery efficiency.

In this context, we designed and synthesized a novel nanocarrier, (CP)6NS, which was found to have good biocompatibility and nucleic acid delivery capability. Therefore, we expect it to be an effective carrier for nucleic acid delivery for tumor therapy.

## 2. Results

### 2.1. Design of (CP)_6_NS

To better realize nucleic acid delivery and tumor treatment, we introduce a new homochiral cyclic peptide containing three aspartates (D) and three prolines (P) connected alternately, named cyclo-(DP)_3_ (Appendix A). Then, we used cyclo-(DP)_3_ as a building block to covalently assemble nanospheres with surface-modifiable 2,2′-diamino-N-methyldiethylamine as the linker. After methylation modification to make the nanospheres positively charged, a potential nanocarrier for nucleic acid delivery was successfully prepared (Figure 1a). The nanospheres prepared based on cyclo-(DP)_3_, namely, (CP)_6_NS, were stable, biocompatible, had low cytotoxicity, and possessed the characteristics of nucleic-acid delivery carriers (Figure 1b) [33]. Compared to other organic materials and cationic polymers [3,34], the significant advantage of (CP)_6_NS was good biocompatibility. Furthermore, (CP)_6_NS could maximally bind and deliver siRNA into tumor cells and cause tumor cell apoptosis. Therefore, we consider (CP)_6_NS to be a potential nanocarrier.

Survivin is a member of the anti-apoptotic protein family (IAPs). It is overexpressed in various cancer cells and is an important therapeutic target. Previous studies showed that inhibiting the overexpression of survivin protein induced apoptosis in tumor cells [35,36,37,38]. Therefore, we used survivin as a potential drug target for further anti-tumor research.

### 2.2. Characterization of (CP)_6_NS

To generate the cyclic peptide nanospheres used for nucleic-acid delivery, we selected appropriate linking motifs and chemically modified them, successfully preparing a potential carrier for nucleic-acid delivery. Nanotubes formed from cyclic peptides can be used for gene transfection [39]. However, most of these cyclic peptide molecules contain histidine or arginine for positively charging the assembly. To make full use of various amino acids, we chose aspartate and proline as the main body of the cyclo-(DP)_3_ cyclic peptide and used functionalized 2,2′-diamino-N-methyldiethylamine as the linker to covalently self-assemble with cyclo-(DP)_3_. Methyl iodide was used to modify the tertiary amine group on the surface of the assembly to endow the assembly with a positive charge.

The fabrication of the assembly was based on a one-step synthesis procedure, and we subsequently performed a series of characterizations of the assembly morphology. The DLS test results showed that the particle size distribution of the assembly was about 90 nm (Appendix A), and the sharp peak shape indicated that the particle size distribution was concentrated, and the particle size was relatively uniform. Relevant studies demonstrated that nanospheres with a small size (<100 nm) had better stability and higher cellular internalization efficiency in vivo [40]. The zeta potential measured by the charge on the surface of the assembly was +17.1 mV (Appendix A). These results verified the feasibility of our design. In addition, SEM and TEM characterization revealed that a large number of solid spherical assemblies, namely, (CP)_6_NS, were formed with clear edges, regular morphology, and good dispersion (Figure 2a–d).

### 2.3. Stability of (CP)_6_NS

Stability is a very important indicator of nano-delivery vehicles. To this end, we detected the size change in (CP)_6_NS in phosphate-buffered saline (PBS) buffer at different times by DLS. We incubated 200 μg/mL of (CP)_6_NS at room temperature for 0 h, 6 h, 12 h, and 24 h, and then detected particle size changes. It was found that the size of (CP)_6_NS did not change significantly within 24 h from the DLS test results (Figure 3a–d), suggesting that (CP)_6_NS nanospheres could remain stable under physiological conditions.

### 2.4. Effect of (CP)_6_NS on Cell Viability

To confirm the cytotoxicity of (CP)_6_NS, an MTT assay was performed. Compared to the control group, the viability of MDA-MB-231 cells (Figure 4a), HCT-116 cells (Figure 4b), and HUVEC cells (Figure 4c) did not change significantly after treatment with different concentrations of (CP)_6_NS up to 500 μg/mL. The viability of all three cell lines was greater than 95%, and the results indicated that (CP)_6_NS had low cytotoxicity in both normal cells and tumor cells. Therefore, we chose 200 μg/mL as the final concentration for subsequent experiments.

To further illustrate the low toxicity characteristics of (CP)_6_NS, toxicity experiments were performed with (CP)_6_NS and the commercially available cationic carriers PLL and PEI. The cell viability of MDA-MB-231 cells did not change significantly after treatment with different concentrations of (CP)_6_NS. However, PLL and PEI treatments significantly reduced cell viability. PEI treatment had a greater impact on cell viability, and the decrease in cell viability reached more than 50% by treatment with different concentrations of PEI (Figure 4d). These results further illustrate the advantage of the lower cytotoxicity of (CP)_6_NS compared to commercially available cationic carriers.

### 2.5. Hemocompatibility of (CP)_6_NS

Hemocompatibility is an important index for evaluating the safety of biological materials, so we evaluated the hemocompatibility of (CP)_6_NS by the hemolysis test. The results showed that after centrifugation, the erythrocytes in normal saline settled at the bottom of the tube, indicating that they were intact. In contrast, erythrocytes in distilled water did not separate after centrifugation due to fragmentation, indicating that hemolysis occurred. After treatment with different concentrations of (CP)_6_NS, the supernatant was clear after centrifugation, indicating that treatment with different concentrations of (CP)_6_NS did not lead to the rupture of erythrocytes (Figure 5a,b).

Referring to relevant standards [41], a hemolysis rate of less than 5% indicates no hemolysis. The measured OD values of different concentrations of (CP)_6_NS and the negative and positive control groups are shown in Appendix A. The OD value of the negative control group (physiological saline) was 0.100, the OD value of the positive control group (distilled water) was 1.439, and the OD values of different concentrations of (CP)_6_NS were 0.110, 0.118, and 0.121. The hemolysis rate of (CP)_6_NS at each concentration was less than 5%, indicating that (CP)_6_NS had good blood compatibility.

### 2.6. The Ability of (CP)_6_NS to Load and Deliver siRNA

The ability of the gene delivery vector to load siRNA is an important indicator used to evaluate its performance. N/P refers to the ratio between the positively charged group carried by the nanocarrier and the negatively charged siRNA phosphate group. An appropriate N/P ratio is key to ensuring gene delivery efficiency. According to relevant literature reports [42] we co-incubated (CP)_6_NS with different N/P ratios (10:1, 25:1, 50:1, 100:1, 120:1) with survivin siRNA. After 4 h, the surface charges of nanospheres under different N/P conditions were detected by the surface Zeta potential of nanospheres with different N/P ratios and were analyzed to evaluate the binding of (CP)_6_NS to siRNA. We fixed the mass of siRNA, gradually increased the mass of the assembly, and calculated the N/P ratio according to the formula. With increases in N/P, the Zeta potential increased from −18.1 mV to +5.6 mV, indicating that siRNA successfully combined with (CP)_6_NS. When the N/P ratio was 100:1, the Zeta potential was +0.66 mV, indicating that almost all of the free siRNA in the solution was bound by (CP)_6_NS (Appendix A). Therefore, an N/P ratio of 100:1 was selected as the binding ratio of (CP)_6_NS to siRNA for the following experiments.

A major obstacle to tumor gene therapy is that the uptake of genes by tumor cells is difficult. Therefore, the cellular uptake ability of gene delivery vectors is also an important indicator for evaluating the performance of vectors. We performed cell imaging by confocal laser microscopy (CLSM) and flow cytometry (FACS) on different sample treatments. In this experiment, the (CP)_6_NS and FAM-siRNA complex at an N/P ratio of 100:1 and an equal amount of free FAM-siRNA were used. The results showed that there was almost no green fluorescence in cells treated with free FAM-siRNA, indicating that free FAM-siRNA had difficulty entering the cells, whereas cells treated with the (CP)_6_NS-FAM-siRNA complex showed obvious green fluorescence, indicating that (CP)_6_NS-FAM-siRNA complexes could be internalized by the cells. Thus, (CP)_6_NS acted as a carrier to facilitate the entry of FAM-siRNA into cells and localization to the cytoplasm (Figure 6a). Subsequently, we used FACS to further quantitatively analyze the intracellular fluorescence intensity (Figure 6b,c), and the results were consistent with the CLSM results.

### 2.7. Anti-Tumor Effects In Vitro

To evaluate the effect of nanocomplexes on the survivin gene, 24 h after transfection with (CP)_6_NS-siRNA complexes, quantitative RT-PCR experiments were conducted to detect changes in the survivin mRNA content in MDA-MB-231 cells. The other experimental groups were the control group (control), the free siRNA group as the negative control, and commercial transfection agent Lipofectamine 2000-siRNA as the positive control (siRNA is survivin siRNA). The level of survivin mRNA in the cells treated with free siRNA did not significantly decrease compared to the control group, whereas the level of survivin mRNA in cells treated with (CP)_6_NS-siRNA complex was significantly decreased by about 42%, and the inhibition rate was comparable to that of the positive control group treated with Lipofectamine 2000-siRNA. No reduction in survivin mRNA levels was observed after treatment with the (CP)_6_NS-scrambled-siRNA complex (Figure 7a). These results indicated that (CP)_6_NS-siRNA successfully down-regulated the level of survivin mRNA in MDA-MB-231 cells, and the actual silencing efficiency was closely related to the nucleic acid sequence of survivin siRNA. Thus, nucleic acid sequences were affected by specific gene silencing.

Subsequently, the changes in survivin protein expression in MDA-MB-231 cells after different treatments were further analyzed by Western blots. After the cells were treated with the (CP)_6_NS-siRNA complex and the Lipofectamine 2000-siRNA complex, the expression levels of survivin protein were significantly reduced (Figure 7b,c). The qRT-PCR and Western blot results showed that the (CP)_6_NS-siRNA complex could effectively reduce the expression levels of survivin mRNA and protein in cells.

The survivin gene, which is overexpressed in various cancer cells, hinders apoptotic programming mainly by inhibiting the activity of apoptotic proteins. Therefore, we delivered survivin siRNA into cancer cells via (CP)_6_NS to inhibit the expression of the survivin gene and detect the apoptosis of cancer cells. The (CP)_6_NS-survivin siRNA complex, the (CP)_6_NS-scramble siRNA complex, and an equal amount of free siRNA were incubated with MDA-MB-231 cells for 24 h, and the apoptosis rate was determined by flow cytometric analysis. The results showed that the apoptosis rate of cells was significantly increased after (CP)_6_NS-survivin siRNA treatment compared to the control group (control). However, (CP)_6_NS-scrambled siRNA and free siRNA treatment had no significant effect on apoptosis (Figure 8a–e). These results indicate that survivin plays a role in inhibiting apoptosis in cells, and further confirm that the (CP)_6_NS-survivin siRNA complex could effectively achieve survivin gene silencing and promote cell apoptosis. Thus, (CP)_6_NS can act as an effective nucleic acid delivery vector for gene therapy.

## 3. Discussion

Peptides have the characteristics of high chemical stability, easy modification, and rich structure, making them one of the most promising nanomaterials. Among them, cyclic peptides, with a flat cyclic structure, have been widely studied by scientists since their discovery because of their structural rigidity and enzymatic stability [43]. Numerous studies have shown that cyclic peptides can form nanotubes, nanofibers, and other structures in a cyclic stacking manner through intermolecular hydrogen bonds [39,44,45]. Therefore, we designed the cyclic peptide cyclo-(DP)_3_ as a novel building block for joining with different linking motifs to expand the assembly strategy of cyclic peptides and construct cyclic peptide-based nucleic acid delivery vehicles. The experimental results showed that the cyclic peptides formed nanospheres whether covalently assembled with 1,6-hexanediamine or electrostatically assembled with cyclic-(RR).

Previous studies reported that the planar rigidity of molecules affected the morphology of the assembly. Generally, the greater the rigidity, the larger the size of the assembly. Other studies showed that the planar rigidity of building blocks is a key factor in the formation of two-dimensional nanovesicles. For example, other research groups used cucurbituril, porphyrin, pillar arene, and other molecules with relatively rigid planes as building blocks. During the assembly process, cross-linking between the building block and the flexible chain first grows laterally to form an oligomeric sheet, and the oligomeric sheet reduces its energy by bending, and then further reacts and grows to form a covalent nanovesicle [46,47]. Cyclic peptides, as cyclic molecules, have unique structural characteristics compared with the above-mentioned macrocyclic molecules. We speculated that the cyclic peptide was affected by intermolecular hydrogen bonds during the assembly process. Due to the simultaneous horizontal and vertical forces, to balance the relationship between the existence of the force and the energy of the aggregate itself, a solid openwork spherical structure is finally presented.

The covalent self-assembly strategy has many unique advantages, such as simple preparation steps, no template, stable structure, and uniform size of the prepared assemblies [48]. Through a rational design, functional nanomaterials with excellent structure and properties can be prepared. Kim et al. used subphthalocyanine and disulfide to prepare high-efficiency antibacterial materials, and then functionally modified the surface of the assembly through the reaction of methyl iodide to give it a positive charge, which increased the generation of nanocapsular reactive oxygen species and antibacterial efficiency [49]. 

In this study, we selected different linker motifs for crosslinking with cyclo-(DP)_3_. First, we selected 1,6-hexanediamine for covalent crosslinking with cyclo-(DP)_3_ (Appendix A). cyclo-(DP)_3_ and 1,6-hexanediamine were covalently cross-linked to form assemblies of a certain size, with an average particle size of about 160 nm and a negative surface charge of −23.3 mV (Appendix A). Electron microscopy characterization showed that the nanospheres were regular in structure and uniform in size, and the interior of the nanospheres was solid (Appendix A). However, their surface potential is not suitable as a nucleic acid delivery vehicle. Therefore, we chose cyclo-(RR) (Appendix A) with a strong positive charge as the linking motif (cyclo-(RR) is a ring structure formed by two arginines connected end to end, and two guanidine groups are provided on both sides of the ring) and cyclo-(DP)_3_ was electrostatically assembled (Appendix A). The assembly formed by cyclo-(DP)_3_ and cyclo-(RR) was similar to the assembly formed by cyclo-(DP)_3_ and 1,6-hexanediamine (Appendix A), and its average particle size was about 150 nm (Appendix A). Although the negative charge on the surface was reduced, it was still not suitable as a nucleic-acid delivery carrier. Finally, we found that positively charging the assembly could be achieved by selecting appropriate linking motifs and functionalizing them, so modifiable 2,2′-diamino-N-methyldiethyl amines were used as linking moieties, and (CP)_6_NS with nanospheres of about 90 nm were successfully constructed based on the covalent self-assembly of cyclo-(DP)_3_ and 2,2′-diamino-N-methyldiethylamine. The amine group was modified by methylation to positively charge the nanospheres, and the nanospheres showed good stability under physiological conditions. Based on these properties, siRNA was loaded by (CP)_6_NS to form (CP)_6_NS-siRNA complexes, which effectively enhanced the cellular uptake of siRNA.

Our results showed that (CP)_6_NS prepared based on cyclo-(DP)_3_ were stable, biocompatible, and had low cytotoxicity. In addition, (CP)_6_NS could maximally bind survivin siRNA and deliver it into MDA-MB-231 cells; that is, (CP)_6_NS had high siRNA loading efficiency. Due to the high positive charge of nanoparticles, they can escape from endosomes through lysosome escape; in addition, because (CP)_6_NS and siRNA are electrostatically bound together, siRNA will gradually release and play a gene interference role with the prolongation of time. Furthermore, after survivin siRNA entered the cells, it significantly decreased the expression levels of survivin mRNA and protein in cells, and further promoted tumor cell apoptosis. Therefore, we believe that (CP)_6_NS is an effective nucleic-acid nanocarrier and has potential applications in tumor therapy.

In general, in this study, cyclo-(DP)_3_ was used as a novel building block, and then covalently cross-linked or electrostatically assembled with different linkers to successfully prepare nanospheres with regular structures, uniform sizes, and good dispersion. We also found that with the prolongation of the electrostatic assembly time, the obtained nanospheres gradually increased in size (Appendix A), indicating the possibility of controlling the assembly size. In addition, using different modifiable linker motifs, such as linker motifs that are responsive to the tumor microenvironment, may also confer targeting capability to the assembly. Therefore, our results suggest a new assembly strategy that extends the current assembly strategy of cyclic peptides.

## 4. Materials and Methods

### 4.1. Reagents and Antibodies

Cyclo-(DP)_3_ was purchased from Shanghai Qiangyao Biological Technology (Shanghai, China). N, N-dimethylformamide (DMF), triethylamine (TEA), and CH3I were obtained from TCI (Shanghai) Development Co., Ltd. (Shanghai, China). Lipofectamine 2000 was provided by Thermo Fisher Scientific (Shanghai, China). Dimethylsulfoxide, hexamethylenediamine, and 2,2′-diamino-N-methyldiethylamine were purchased from Solarbio Life Sciences (Beijing, China). Survivin-siRNA, FAM-siRNA, and scrambled siRNA were purchased from GenePharma (Shanghai, China). Primers were purchased from Comate Bioscience (Changchun China). MTT was obtained from Genview (Beijing, China). DAPI, TRIzol reagent, and sodium dodecyl sulfate-polyacrylamide gel electrophoresis (SDS-PAGE) gel preparation kits were acquired from Solarbio Life Sciences (Beijing, China). All-In-One RT MasterMix kits were purchased from TransGen Biotech (Beijing, China). The UNIQ-10 Column Total RNA Extraction Kit, Go Taq qPCR Master Mix Kit, and Annexin V-FITC/PI Apoptosis Detection Kit were purchased from Sangon Biotech (Shanghai,China). Anti-survivin antibody A5719 was obtained from Bimake (Houston, TX, USA). GAPDH (rabbit, AP0066) was obtained from Bioworld (Minneapolis, MN, USA).

### 4.2. Preparation of Nanospheres by Reaction of Cyclo-(DP)_3_ and 1,6-Hexanediamine

Nanospheres were prepared by dissolving 0.50 mg (0.000786 mmol) of cyclo-(DP)_3_ and 0.41 mg (0.00356 mmol) of 1,6-hexanediamine in 2 mL of DMF. The molar ratio of the reactive functional carboxyl and amino groups in the system was exactly 1:3. Then, 3.67 mg of PyBOP and 4.89 μL of TEA were added with stirring, and the reaction was carried out at room temperature for 24 h. Dialysis was performed to obtain the assembled nanospheres.

### 4.3. Preparation of Nanospheres by Reaction of Cyclo-(DP)_3_ and Cyclo-(RR)

Nanospheres were prepared by dissolving 0.50 mg (0.000786 mmol) of cyclo-(DP)_3_ and 1.83 mg (0.0059 mmol) of cyclic-(RR) in 2 mL of DMF at a molar ratio of the functional carboxyl to guanidine groups in the system of exactly 1:5. The mixture was stirred at room temperature for 24 h. Dialysis was performed to obtain the assembled nanospheres.

### 4.4. Construction of (CP)_6_NS

To construct (CP)_6_NS, 4 mg (0.006288 mmol) of cyclo-(DP)_3_ and 3.6 μL (0.02832 mmol) of 2,2′-diamino-N-methyldiethylamine were dissolved in 12 mL of DMF. The molar ratio was exactly 1:3. Then, 29.36 mg of PyBOP and 32 μL of TEA were added under stirring conditions, fully reacted for 24 h, dialyzed, and methylated by CH_3_I. The reaction mixture was dialyzed to obtain positively charged nanospheres.

### 4.5. Characterization of the Assembly Structure

Dynamic light scattering characterization (DLS): 1 mL of the assembly solution was placed in the size and Zeta sample cells and the particle size and surface potential of the assembly were measured.

Scanning electron microscopy characterization (SEM): A silicon wafer was hydroxylated in advance, then dried with nitrogen, and placed on a clean weighing paper. The assembly solution (10 μL) on the silicon wafer was dried at room temperature and analyzed by gold spraying.

Transmission electron microscopy (TEM): The copper mesh was removed and placed on a hydrophobic weighing paper. The assembly solution (5 μL) was dropped vertically onto the copper mesh and dried at room temperature.

### 4.6. Cell Culture and MTT Assays

MDA-MB-231, HCT-116, and HUVEC cells were cultured in DMEM with 10% FBS at 37 °C with 5% CO_2_. For MTT assays, 1 × 10^4^ cells/well were seeded in 96-well plates. After 24 h, the drug was added and incubated for another 24 h, followed by the addition of MTT and detection of formazan at 450 nm.

### 4.7. Hemolysis Test

(CP)_6_NS at final concentrations of 20, 200, and 500 μg/mL were prepared in normal saline for use. A 2 mL sample of blood was taken from healthy SD rats, and the broken red blood cells were removed by centrifugation. Freshly diluted blood (150 μL) was added to 1 mL of the sample at each concentration, shaken gently at 37 °C for 90 min, centrifuged, and the absorbance of the supernatant in each group was measured at 545 nm. The hemolysis rate was calculated by the following formula:$$\mathrm{Hemolysis}\;\mathrm{rate}\;(\%)=\frac{{\mathrm{OD}}_{\mathrm{sample}}{\;-\;\mathrm{OD}}_{\mathrm{negative}\;\mathrm{control}}}{{\mathrm{OD}}_{\mathrm{positive}\;\mathrm{control}}{-\;\mathrm{OD}}_{\mathrm{negative}\;\mathrm{control}}}\times 100\%$$

### 4.8. (CP)_6_NS and siRNA Binding Ability Assay

(CP)_6_NS was dissolved to 500 μg/mL in DEPC water, and siRNA was used at a concentration of 20 μM. (CP)_6_NS with different N:P ratios (10, 25, 50, 100, 120) were incubated with siRNA for 4 h, and the surface charges on the nanospheres were detected by DLS.

### 4.9. Detection of (CP)_6_NS Ability to Deliver FAM-siRNA

CLSM detection: MDA-MB-231 cells were plated in confocal culture dishes at a density of 2 × 10^5^ cells/dish. The (CP)_6_NS and FAM-siRNA complex with an N/P ratio of 100:1 and a final concentration of 200 μg/mL of (CP)6NS was used as the test group. The negative control group contained only the same amount of FAM-siRNA. The final concentration of siRNA was 100 nM. After treatment for 4 h, the culture medium was aspirated, washed, fixed with cell fixative (4% paraformaldehyde) at 4 °C for 20 min, stained with DAPI nuclear dye for 10 min, and washed with PBS. The fluorescence intensity was observed by the LSM-710 (ZEISS, Berlin, Germany) inverted fluorescence microscope.

FACS detection: The cell treatment method was the same as for CLSM. After 4 h of treatment, the medium in the wells was removed, and the cells were washed once with PBS, digested, and centrifuged. The cell pellet was resuspended in PBS, and the supernatant was discarded following centrifugation. Finally, the collected cells were resuspended in 500 μL of PBS and filtered through a 200-mesh filter into a flow tube, and the fluorescence intensity in the cells was analyzed by flow cytometry.

### 4.10. Quantitative Real-Time-PCR and Western Blot Assays

MDA-MB-231 cells were seeded in a 6-well plate at a cell density of 5 × 10^5^ cells/well in each well. The cells were cultured for 12 h to allow for adherence, treated with the indicated drugs, and incubated for 24 h. The Total RNA Extraction Kit was used to extract total RNA from the cells, and then reverse transcription and qPCR were performed using the All-In-One RT Master Mix kit and Go Taq qRT-PCR Master Mix kit. qRT-PCR was performed with the primer pairs in Appendix A. The proteins were extracted with a protein extraction kit. After measuring the protein concentration with the BCA kit, SDS-PAGE was performed and the proteins were transferred to polyvinylidene fluoride (PVDF) membranes. The membranes were incubated with primary antibodies and secondary antibodies, and the reactions were visualized using the Tanon 5200 (Tanon Science & Technology Co., Ltd., Shanghai, China) imaging system.

### 4.11. Cell Apoptosis

The MDA-MB-231 cells were seeded into a 6-well plate and incubated with different samples for 24 h. Then, the cells were digested with trypsin without EDTA, collected, and washed twice with cold PBS. The cells were stained using the Annexin V-FITC/PI Kit, and the apoptosis rate in each treatment group was detected by flow cytometry.

### 4.12. Statistical Analyses

GraphPad Prism 7 software was used for statistical analyses of the obtained data. The data were quantitatively expressed as the mean ± SD value (*** *p* < 0.001, ** *p* < 0.01, * *p* < 0.05). Statistical comparisons between two groups were performed using the unpaired two-tailed *t*-test. For all results, a *p*-value of < 0.05 was considered statistically significant.

## 5. Conclusions

Based on the covalent self-assembly of cyclo-(DP)_3_ and 2,2′-diamino-N-methyldiethylamine, the (CP)_6_NS nanocarrier was successfully constructed for nucleic acid delivery. It was stable, biocompatible, and had low cytotoxicity. It was shown to deliver survivin siRNA into cells, down-regulate the expression of the survivin gene, and promote the apoptosis of tumor cells. Since the vector could transfect survivin siRNA into cells and generate gene interference without the aid of polymers or liposomes, we consider (CP)_6_NS to be a promising nanocarrier.

## Figures and Tables

**Figure 1 ijms-23-12071-f001:**
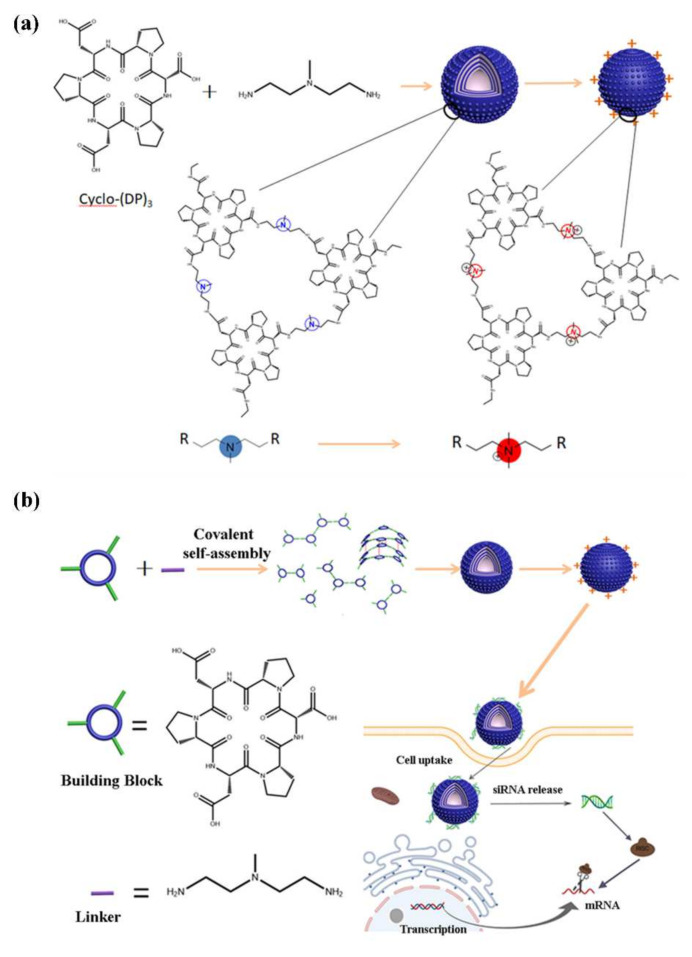
Schematic representation of cyclic peptide nanospheres for siRNA delivery. (**a**) Schematic diagram of the self-assembly of cyclo-(DP)_3_, and 2,2′-diamino-N-methyldiethylamine and its methylation modification. (**b**) Schematic diagram of the design of (CP)_6_NS and the siRNA delivery for gene therapy.

**Figure 2 ijms-23-12071-f002:**
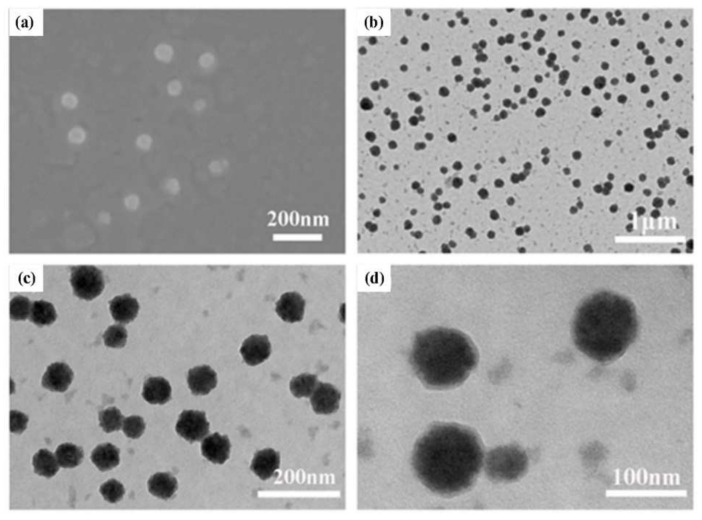
Characterization of (CP)_6_NS. (**a**) Scanning electron microscopy image of (CP)_6_NS. (**b**–**d**) TEM image of (CP)_6_NS.

**Figure 3 ijms-23-12071-f003:**
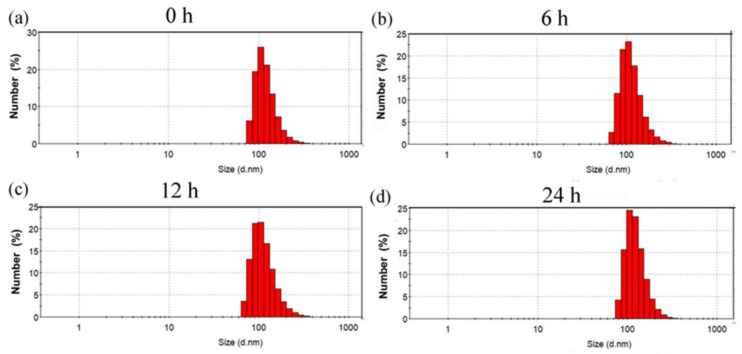
Particle size changes in (CP)_6_NS at room temperature for (**a**) 0 h, (**b**) 6 h, (**c**) 12 h, and (**d**) 24 h.

**Figure 4 ijms-23-12071-f004:**
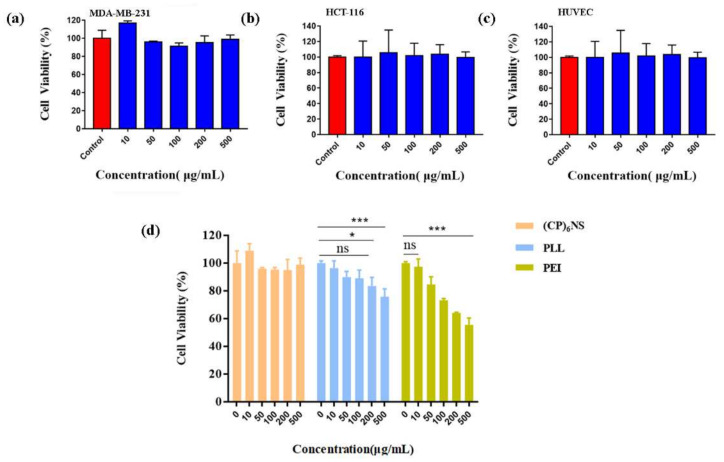
The MTT assay was conducted with different cells. (**a**) MDA-MB-231 cells, (**b**) HCT-116 cells, and (**c**) HUVEC cells. (**d**) (CP)_6_NS were compared to PLL and PEI. MTT assays were used to assess their biocompatibility. Each value is expressed as the mean ± SD (*n* = 3).

**Figure 5 ijms-23-12071-f005:**
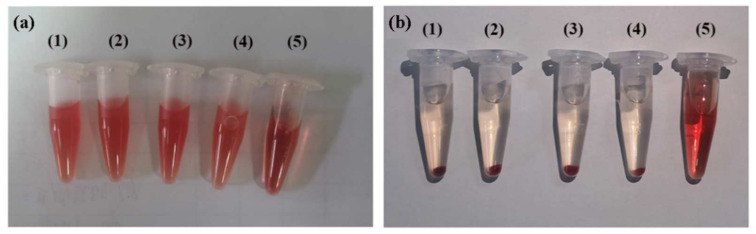
Hemolysis assay. Blood from Sprague–Dawley (SD) rats was centrifuged and washed to obtain red blood cells, incubated with different concentrations of (CP)_6_NS, and centrifuged. (**a**) Before centrifugation and (**b**) after centrifugation. (1) Incubation with (CP)_6_NS at 20 μg/mL, (2) (CP)_6_NS at 200 μg/mL, (3) (CP)_6_NS at 500 μg/mL, (4) saline, and (5) water.

**Figure 6 ijms-23-12071-f006:**
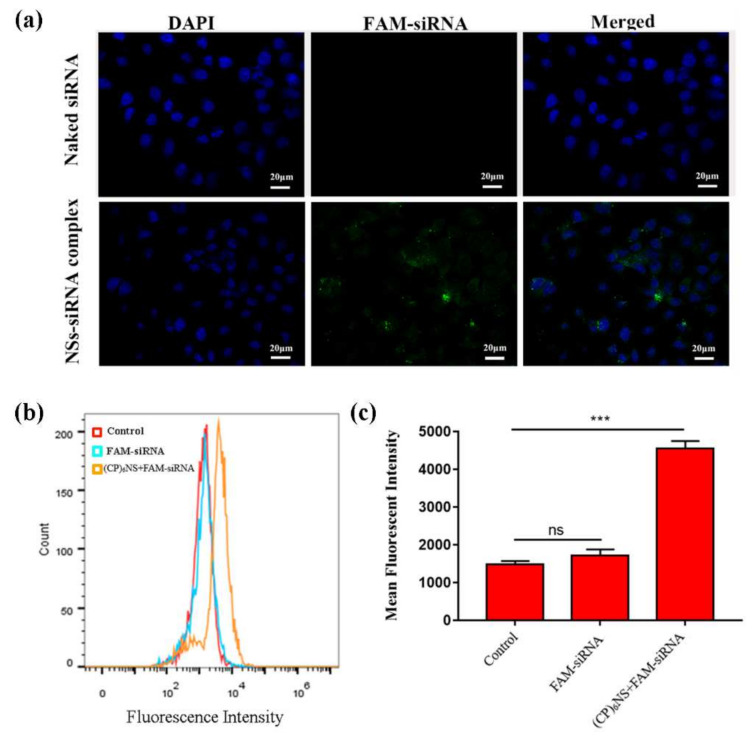
Internalization of (CP)_6_NS-siRNA into MDA-MB-231 cells. (**a**) CLSM images of MDA-MB-231 cells incubated with free FAM-siRNA and (CP)_6_NS+FAM-siRNA complexes for 4 h. (**b**) FACS analysis table. (**c**) Mean fluorescence intensity statistics. Each value is expressed as the mean ± SD (*n* = 3).

**Figure 7 ijms-23-12071-f007:**
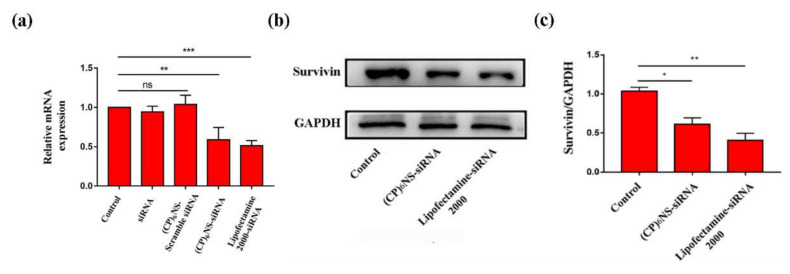
Effect on survivin by (CP)_6_NS delivering survivin siRNA. (**a**) The expression of survivin mRNA was analyzed by qRT-PCR. (**b**) The expression of survivin protein was analyzed by Western blots. (**c**) Statistics on the expression of survivin protein with GAPDH as the internal reference. Each value is expressed as the mean ± SD (*n* = 3).

**Figure 8 ijms-23-12071-f008:**
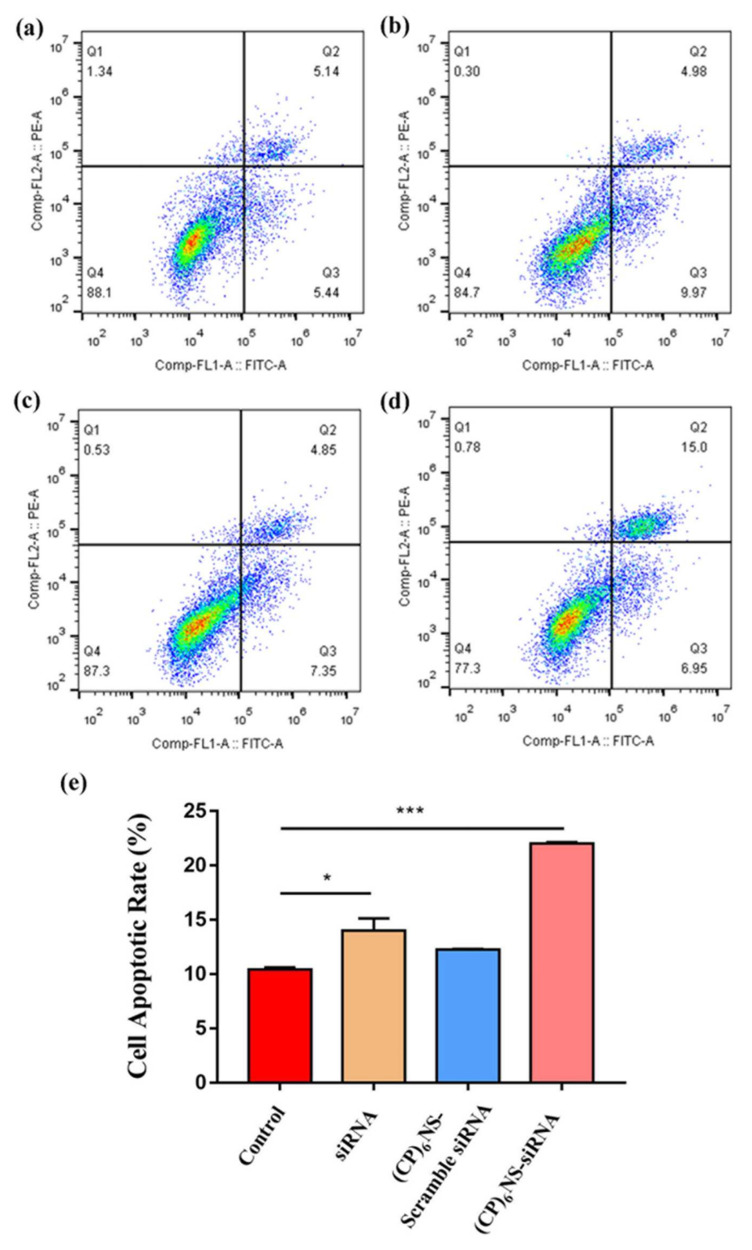
Flow cytometric analysis of cell apoptosis in each treatment group. (**a**) Control group, (**b**) free siRNA group, (**c**) (CP)_6_NS-scrambled siRNA treatment group and (**d**) (CP)_6_NS-survivin siRNA treatment group. (**e**) Statistical analysis of apoptosis rates. Each value is expressed as the mean ± SD (*n* = 3).

## Data Availability

Not applicable.

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
