# Peer review of "Design of Cyclic Peptide-Based Nanospheres and the Delivery of siRNA"

_ijms, 2022, doi:10.3390/ijms232012071_

Round 1

Reviewer 1 Report

The authors report the development of peptide-based nanospheres for siRNA delivery and their characterization and in vitro evaluation. The purpose of the article is innovative. Extensive work was done, with interesting results being moved to the supplementary material. There is coherence in the methodology, however, questions should be addressed before publication: 

1.      Introduction: The 5th and 6th paragraphs include results and seem to be more suitable for the abstract/results then introduction. They should be replaced by a paragraph explaining the choices of the authors, to connect everything into a clear objective of the work, which is innovative and interesting. 

2.      Introduction: Figure S1 shows chromatograms for the identification of cyclo-(DP)3. Is the citation Figure S4?

3.      The authors introduced a new homochiral cyclo-(DP)3. How do the authors synthesized this novel material? The authors should either cite a previous work or show the chemical pathway (maybe in the supplementary material). The peaks in Figure S1 should also be identified and related to the structure of cyclo-(DP)3.

4.      Topic 2.2: The stability would be better understood when analyzing DLS graphs by intensity, not by numbers. Could you please provide the histogram by intensity? Moreover, did the authors check the stability after siRNA binding to the surface?  

5.      Topic 2.3: Have the authors checked the cytotoxicity after the siRNA binding? This will be necessary information to understand the right time to perform PCR assay so the silencing is detected due to the RNAi effect, and not cell death. 

6.      Topic 2.4:  Table 3.7 or Table S1?

7.      Topic 2.5: The entry pathway of nanoparticles is often by endocytosis. Is the siRNA in the cytosol, or inside endosome/lysosome? How do the authors expect the particle to escape endosomes?

8.      How do the authors expect the release of siRNA from the nanostructure?

9.      Topic 2.6: The authors showed survivin gene knockdown in 24 hours, with full integrity of GAPDH expression. However, the authors show significant cell apoptosis in the same 24 hours of treatment. Cell death decreases all the genes ubiquitously. Is there a probability that the survivin silencing may be due to cell apoptosis rather than RNAi effect? Maybe a timely cytotoxicity assay could clarify this question.

10.  Methodology: The authors should include a different method to quantify the encapsulation efficiency, rather than the estimation by Zeta potential.  

Author Response

Response to Reviewer 1 Comments

Thanks for your letter.

Thanks for your critical examination. Your comments are very helpful for the revision of this manuscript. We are sending the revised manuscript according to the comments. The revised parts are in red and the reply to each point is listed below:

  1. 1. Introduction: The 5th and 6th paragraphs include results and seem to be more suitable for the abstract/results then introduction. They should be replaced by a paragraph explaining the choices of the authors, to connect everything into a clear objective of the work, which is innovative and interesting.

Response: Thanks for your suggestion. According to your suggestion, we have revised Figure 1 to Scheme 1, because this figure is the design idea of the nanoparticle, so we named it Scheme 1, which may better reflect our innovation.

  1. 2. Introduction: Figure S1 shows chromatograms for the identification of cyclo-(DP)3. Is the citation Figure S4?

Response: Thanks for your suggestion. Figure S1 shows the chromatogram used to identify cyclo-(DP)3, shown in both Figure 1 and citation Figure S4.

  1. 3. The authors introduced a new homochiral cyclo-(DP)3. How do the authors synthesized this novel material? The authors should either cite a previous work or show the chemical pathway (maybe in the supplementary material). The peaks in Figure S1 should also be identified and related to the structure of cyclo-(DP)3.

Response: Thanks for the valuable information from the reviewer. The homochiral cyclo-(DP)3 in this paper was synthesized by a company we commissioned. The molecular weight in Figure S1 is consistent with the molecular weight of cyclo-(DP)3, so we think it is cyclo-(DP)3 synthesized.

  1. 4. Topic 2.2: The stability would be better understood when analyzing DLS graphs by intensity, not by numbers. Could you please provide the histogram by intensity? Moreover, did the authors check the stability after siRNA binding to the surface?

Response: Thanks for your suggestion. Based on your comments, we have added histograms to the article. In addition, the stability of siRNA binding to the surface was not examined in this paper and may be more refined based on your suggestions in the subsequent optimization of (CP)6NS.

  1. 5. Topic 2.3: Have the authors checked the cytotoxicity after the siRNA binding? This will be necessary information to understand the right time to perform PCR assay so the silencing is detected due to the RNAi effect, and not cell death.

Response: Thanks for the valuable information from the reviewer. In Figure 7a, we introduced scrambled-siRNA and found that the intracellular survivin mRNA level did not change. However, both (CP)6NS and the marketed transfection kit can cause a significant decrease in the level of survivin mRNA, so we believe that gene silencing does occur.

  1. 6. Topic 2.4: Table 3.7 or Table S1?

Response: Thanks for your suggestion. This may be a previous manuscript version, which we have now revised.

  1. 7. Topic 2.5: The entry pathway of nanoparticles is often by endocytosis. Is the siRNA in the cytosol, or inside endosome/lysosome? How do the authors expect the particle to escape endosomes?

Response: Thanks for your suggestion. Since our synthesized nanoparticles have a high positive charge, we expect them to escape endosomes through the lysosomal escape process.

  1. 8. How do the authors expect the release of siRNA from the nanostructure?

Response: Thanks for the valuable information from the reviewer. Since the binding of our synthesized nanoparticles to siRNA is electrostatic adsorption, the binding force may gradually decrease with time, which will eventually lead to the release of the siRNA adsorbed on the nanoparticles.

  1. 9. Topic 2.6: The authors showed survivin gene knockdown in 24 hours, with full integrity of GAPDH expression. However, the authors show significant cell apoptosis in the same 24 hours of treatment. Cell death decreases all the genes ubiquitously. Is there a probability that the survivin silencing may be due to cell apoptosis rather than RNAi effect? Maybe a timely cytotoxicity assay could clarify this question.

Response: Thanks for your suggestion. Although apoptosis can lead to a decrease in the level of certain genes, the expression of the internal reference protein GAPDH generally does not change significantly (such as detecting apoptotic proteins), and we are performing WB to ensure that the concentration of the loading protein is the same, so we believed that the decrease in survivin protein is indeed due to the knockout of survivin gene. For gene silencing, the introduction of scrambled-siRNA did not lead to the decrease of survivin gene, while the introduction of complete siRNA led to the decrease of survivin gene expression, so we thought that gene silencing occurred.

  1. 10. Methodology: The authors should include a different method to quantify the encapsulation efficiency, rather than the estimation by Zeta potential.

Response: Thanks for the valuable information from the reviewer. At first we used nucleic acid electrophoresis to quantify the encapsulation efficiency, but since siRNA is adsorbed on the surface of nanoparticles due to electrostatic interaction, the quantification of nucleic acid electrophoresis cannot explain its encapsulation efficiency, and finally we estimated its encapsulation efficiency by Zeta potential .

Reviewer 2 Report

Dear Editor, I have read with interest the manuscript “Design of Cyclic Peptide-based Nanospheres and the Delivery of siRNA ”, by Dr. Liu, Dr. Guan and collaborators, on the possible application of peptide-based nanoparticles for siRNA delivery. The study refers to potential ‘drug-delivery’-like application of this new type of nanoparticles. The Authors have shown that this sort of nanoparticles can bind siRNA, allow intracellular penetration, and that siRNA decreases the mRNA level of an anti-apoptosis protein. They also measured the amount of protein, which result reduced after treatment in cultured cells. The cell apoptosis was also measured, resulting in the (desired) increase. The study is overall quite convincing, data are clear and all ‘necessary’ steps have been done. In a sense, this is a quite ‘standard’ study, applied on a new class of nanocarriers. So, no exciting new measurements have been introduced. All is based on solid robust traditional approaches. In conclusion, I believe that this manuscript can be published in IJMS after the modifications indicated below, in order to improve the manuscript quality and reader understanding. List of comments, issues, criticisms. Numbers refer to line numbers ---------------------------------------------------------------------------------- 31: which siRNA? Please specify. 31: In which sense “about”? The length should be exactly known 32: “degrades”? maybe “induces degradation”. siRNA has no enzymatic digestive actitivity 65: referring to Figure S1: I cannot read the m/z values in figure. Please add notes in the Figure S1 caption 66: add information about the role as ‘linker’ for 2,2’-…., as done in the abstract 72: Authors make the claim “Compared to other organic materials and cationic polymers [34,35], the significant advantage of (CP)6NS was good biocompatibility and biodegradability”, however it is not clear what are the experiments that led to this conclusion. Please explain, demonstrate, or delete the sentence. 99: the sentence is wrong: “the particle size distribution … was about 91.28”. Maybe the Authors refer to the mean size or other analogous measure. Moreover, if it is ‘about’ why inserting two figures after the decimal point? It has no sense. Maybe write “was about 90 nm”, and specify 91.3 nm in other place (e.g. figure caption). Also, I do not believe that DLS instrument can have a precision of 0.01 nm (0.1 Angstrom!). That number (91.28) is just a result of averaging math operations. 100: reference to Figure S6 is done. However, in the text there was no reference to Figures S2-S5. Please number S-figures in consecutive way as they appear on the main text. Perhaps the Authors mean Figure S2? Actually Figure S2 shows the size distribution ranging from 60 to 200 nm. This confirms my comment that the 0.28 part of the 91.28 number is just a result of arithmetic averaging. 114: why did the Authors select exactly the concentration of 200 ug/mL? Is this physiologically relevant? What is the expected concentration of future usage? What happens at higher concentrations? 125: similar question for the MTT test? Why did the Authors select the 500 ug/mL? By the way, if the nanoparticle usage is based on 500 ug/mL, why the physical stability test was performed at lower concentration (200 ug/mL)? 152: The table is named in strange way (Table 3.7). Is this referring to a previous version of this manuscript, maybe as book chapter? Please clarify. 164: maybe P/N corresponds to the explanation in line 164-165 (not N/P) (??). Either change the ratio or the explanation. Check also all further occurrences of N/P ratio. Maybe N means nitrogen and P phosphorous, but incidentally N also means negative and P positive. It is now a bit confusing. 172: I am confused about the explanation. If the ratio siRNA/methylated peptide increases, the zeta potential should decrease. Am I right? Please check 207: please explain here what does it mean: “ (CP)6NS-scrambled-siRNA complex”

Author Response

Response to Reviewer 2 Comments

Thanks for your letter.

Thanks for your critical examination. Your comments are very helpful for the revision of this manuscript. We are sending the revised manuscript according to the comments. The revised parts are in red and the reply to each point is listed below:

  1. 1. 31: which siRNA? Please specify. 31: In which sense “about”? The length should be exactly known.

Response: Thanks for your suggestion. Here we introduce the concept of siRNA for gene therapy. Here, siRNA refers to a type of RNA used for gene therapy, so its name and length do not refer to a specific siRNA.

  1. 32: “degrades”? maybe “induces degradation”. siRNA has no enzymatic digestive actitivity.

Response: Thanks for the valuable information from the reviewer. Based on your suggestions, we have carefully revised the article.

  1. 3. 65: referring to Figure S1: I cannot read the m/z values in figure. Please add notes in the Figure S1 caption.

Response: Thanks for your suggestion. According to your suggestion, the enlarged view of Figure S1 is as follows.

  1. 4. 66: add information about the role as ‘linker’ for 2,2’-…., as done in the abstract.

Response: Thanks for your suggestion. Based on your suggestions, we have carefully revised the article.

  1. 5. 72: Authors make the claim “Compared to other organic materials and cationic polymers [34,35], the significant advantage of (CP)6NS was good biocompatibility and biodegradability”, however it is not clear what are the experiments that led to this conclusion. Please explain, demonstrate, or delete the sentence.

Response: Thanks for your suggestion. Following your suggestion, we removed the term biodegradability, and biocompatibility is shown in Figure 4d.

  1. 6. 99: the sentence is wrong: “the particle size distribution … was about 91.28”. Maybe the Authors refer to the mean size or other analogous measure. Moreover, if it is ‘about’ why inserting two figures after the decimal point? It has no sense. Maybe write “was about 90 nm”, and specify 91.3 nm in other place (e.g. figure caption). Also, I do not believe that DLS instrument can have a precision of 0.01 nm (0.1 Angstrom!). That number (91.28) is just a result of averaging math operations.

Response: Many thanks to the reviewers for the valuable information. We have made careful revisions here based on your suggestions.

  1. 7. 100: reference to Figure S6 is done. However, in the text there was no reference to Figures S2-S5. Please number S-figures in consecutive way as they appear on the main text. Perhaps the Authors mean Figure S2? Actually Figure S2 shows the size distribution ranging from 60 to 200 nm. This confirms my comment that the 0.28 part of the 91.28 number is just a result of arithmetic averaging.

Response: Thanks for your suggestion. We have made careful revisions here based on your suggestions. 100: Figure S2; 175: Figure S3; 276: Figure S4; 279: Figure S5. And for the size distribution of nanoparticles, this is really our neglect, thanks again for the suggestion.

  1. 8. 114: why did the Authors select exactly the concentration of 200 ug/mL? Is this physiologically relevant? What is the expected concentration of future usage? What happens at higher concentrations?

Response: Thanks for the valuable information from the reviewer. Since the concentration of (CP)6NS is 200 ug/mL at the optimal N/P ratio, we chose the concentration of 200 ug/mL for the experiments.

  1. 9. 125: similar question for the MTT test? Why did the Authors select the 500 ug/mL? By the way, if the nanoparticle usage is based on 500 ug/mL, why the physical stability test was performed at lower concentration (200 ug/mL)?

Response: Thanks for the valuable information from the reviewer. Since the concentration of (CP)6NS is 200 ug/mL at the optimal N/P ratio, we chose the concentration of 200 ug/mL for the experiments. And MTT is to detect the toxicity of (CP)6NS, so we set the highest concentration of 500 ug/mL.

  1. 10. 152: The table is named in strange way (Table 3.7). Is this referring to a previous version of this manuscript, maybe as book chapter? Please clarify.

Response: Thanks for your suggestion. This may be a previous manuscript version, which we have now revised.

  1. 11. 164: maybe P/N corresponds to the explanation in line 164-165 (not N/P) (??). Either change the ratio or the explanation. Check also all further occurrences of N/P ratio. Maybe N means nitrogen and P phosphorous, but incidentally N also means negative and P positive. It is now a bit confusing.

Response: Thanks for your suggestion. When using charged materials to synthesize nanoparticles, the ratio of positive charge to negative charge will affect the stability, potential and other properties of nanoparticles. As in nucleic acid lipid nanoparticles, the positive charge is usually a cationic lipid with ionizable ammonium (N), while the negative charge is a nucleic acid molecule with a large amount of phosphate (P), which can be combined by electrostatic adsorption. Therefore, the unreasonable proportion may lead to the particle size is too large, poor stability and other defects, the correct calculation of the positive charge and negative charge ratio - N/P ratio - is particularly important.

  1. 12. 172: I am confused about the explanation. If the ratio siRNA/methylated peptide increases, the zeta potential should decrease. Am I right?

Response: Thanks for the valuable information from the reviewer. Indeed, as the siRNA/nanoparticle ratio increases, the zeta potential decreases. Since nanoparticles are positively charged and nucleic acids are negatively charged, the adsorption of the two leads to a change in zeta potential.

  1. 13. 207: please explain here what does it mean: “(CP)6NS-scrambled-siRNA complex”

Response: Thanks for the valuable information from the reviewer. Here, in addition to survivin-loaded siRNA, we also loaded scrambled-siRNA as a control.
